# The Establishment and Optimization of a Chicken Primordial Germ Cell Induction Model Using Small-Molecule Compounds

**DOI:** 10.3390/ani14020302

**Published:** 2024-01-18

**Authors:** Wei Gong, Juanjuan Zhao, Zeling Yao, Yani Zhang, Yingjie Niu, Kai Jin, Bichun Li, Qisheng Zuo

**Affiliations:** 1Joint International Research Laboratory of Agriculture and Agri-Product Safety of Ministry of Education of China, Yangzhou University, Yangzhou 225009, China; mx120220894@stu.yzu.edu.cn (W.G.); 13351187421@163.com (J.Z.); 211603122@stu.yzu.edu.cn (Z.Y.); ynzhang@yzu.edu.cn (Y.Z.); niuyj@yzu.edu.cn (Y.N.); 007838@yzu.edu.cn (K.J.); yubcli@yzu.edu.cn (B.L.); 2Key Laboratory of Animal Breeding Reproduction and Molecular Design for Jiangsu Province, College of Animal Science and Technology, Yangzhou University, Yangzhou 225009, China

**Keywords:** chicken, primordial germ cells, embryonic stem cells, RNA-seq, BMP4

## Abstract

**Simple Summary:**

In this study, we screened key signaling pathways regulating the formation of chicken PGCs using transcriptome sequencing and established a two-step induction model of PGCs in vitro to improve the optimization of their induction efficiency. Our results provide cellular materials for realizing the specific application of PGCs.

**Abstract:**

In recent years, inducing pluripotent stem cells to differentiate into functional primordial germ cells (PGCs) in vitro has become an important method of obtaining a large number of PGCs. However, the instability and low induction efficiency of the in vitro PGC induction system restrict the application of PGCs in transgenic animal production, germplasm resource conservation and other fields. In this study, we successfully established a two-step induction model of chicken PGCs in vitro, which significantly improved the formation efficiency of PGC-like cells (PGCLCs). To further improve the PGC formation efficiency in vitro, 5025 differentially expressed genes (DEGs) were obtained between embryonic stem cells (ESCs) and PGCs through RNA-seq. GO and KEGG enrichment analysis revealed that signaling pathways such as BMP4, Wnt and Notch were significantly activated during PGC formation, similar to other species. In addition, we noted that cAMP was activated during PGC formation, while MAPK was suppressed. Based on the results of our analysis, we found that the PGC formation efficiency was significantly improved after activating Wnt and inhibiting MAPK, and was lower than after activating cAMP. To sum up, in this study, we successfully established a two-step induction model of chicken PGCs in vitro with high PGC formation efficiency, which lays a theoretical foundation for further demonstrating the regulatory mechanism of PGCs and realizing their specific applications.

## 1. Introduction

In recent years, primordial germ cells (PGCs) have been widely used in the fields of genetic engineering, regenerative medicine and germplasm conservation due to their unique biological characteristics. Applications have been carried out for a number of species, including mice, zebrafish and *Drosophila* [1]. However, these applications have not been widely conducted or realized in poultry due to the fact that the developmental process of avian PGCs, the characteristics of which are specific, differs greatly from those of other species [2]. What is important is that, as a result of the correlational studies being carried out relatively recently, the developmental regulatory mechanisms of avian PGCs have not been fully elucidated, which makes the isolation and culture system in vitro for avian PGCs unstable and makes it difficult to obtain large quantities of PGCs for specific applications. In particular, it is more difficult to isolate PGCs from some endangered poultry in vivo. It is well known that embryonic stem cells (ESCs), as pluripotent stem cells, are capable of differentiating into any type of cell, so inducing ESCs to differentiate into PGCs in vitro provides us with a new way to obtain PGCs. Compared with mammals, the long-term culture and identification of chicken ESCs is difficult due to species difference. However, Zhang et al. [3] successfully obtained somatic and gonadal chimeric individuals by culturing isolated cells from chicken blastocysts and injecting them back into recipient chicken embryos, which solved the problem of the source of ESCs during the in vitro induction of chicken PGCs.

In fact, the system of inducing ESCs into PGC-like cells (PGCLCs) in vitro has been relatively well established in mice, humans, pigs and other species [4]. For example, Toyooka et al. [5] successfully induced ESCs to PGCLCs by BMP4, with an efficiency of 2.9% ± 0.7%; Hayashi et al. [6] added activin A, BMP8b and other small-molecule compounds on this basis, increasing the PGCLC induction efficiency to more than 40%. However, due to the uniqueness of PGC development in poultry and the uncertainty of its regulatory mechanism, the current in vitro PGC induction system in poultry (chicken) is relatively isolated and its efficiency is relatively low. For example, Li’s team induced chicken ESCs into PGCLCs by using RA and BMP4, respectively, but the efficiency was less than 10% [7,8]. Although Lu’s team [9] successfully induced pluripotent stem cells into germ cells expressing *Dazl*, *c-kit* and other marker genes, the cell type was challenging to identify. It is not difficult to realize that the in vitro induction system of poultry PGCs is not perfect, and these studies show that the system is not well established.

In mammals (taking mice or humans as examples), in order to improve the PGC induction efficiency in vitro, researchers conducted a systematic study on the in vivo and in vitro PGC generation process and found that ESCs need to undergo an embryoid body or epiblast state during the induction and differentiation of PGCs in vitro before further induction and differentiation into PGCLCs. Therefore, Hayashi et al. [6] established a two-step PGC induction model, in which ESCs were first induced into embryoid bodies or epiblasts by inducers including RA; then, PGCLCs were induced by BMP4 and Activin A, with an induction efficiency of more than 50%. In previous studies, Zuo et al. [10] also found that embryoid bodies also appeared in the process of germ cell formation induced by RA or BMP4 in vitro. Therefore, the two-step induction model can also be adopted in the process of inducting pluripotent stem cells into PGCLCs in vitro in chickens.

In order to establish a two-step induction model of chicken PGCs in vitro, this study intends to systematically screen the key factors in the chicken PGC formation process using RNA sequencing; the aim was to optimize the established two-step induction model with appropriate activators and inhibitors to systematically improve the PGC induction efficiency, providing materials for the further application of avian PGCs.

## 2. Materials and Methods

### 2.1. Ethics Statement

All of the procedures involving the care and use of animals conformed to the U.S. National Institute of Health guidelines (NIH Pub. No. 85-23, revised 1996) and were approved by the Laboratory Animal Management and Experimental Animal Ethics Committee of Yangzhou University. The fertilized eggs of Rugao yellow chickens were purchased from the Poultry Research Institute of the Chinese Academy of Agricultural Sciences and hatched under an environment of 37 °C with 70% humidity.

### 2.2. Cell Isolation and Culture

The freshly fertilized eggs from Rugao yellow chickens were provided by the Poultry Research Institute of CAAS, Yangzhou, Jiangsu Province, China. Blastoderm cells from the embryonic region of the X-stage fertilized eggs were used to isolate ESCs. The isolated blastoderm cells were cultured in KO-DMEM (Gibco, Carlsbad, CA, USA, 10829018), containing 10% KSR (Gibco, 10828028), 2.0 mM GlutaMax (Gibco, 35050061), 1% non-essential amino acids (Gibco, 11140050), 1% chicken serum (Gibco, 16110082), 0.1 mM β-mercaptoethanol (Gibco, 21985023), 10 ng/mL LIF (Sigma-Aldrich, St. Louis, MO, USA, ESG1106), 10 ng/mL bFGF (Sigma-Aldrich, GF446), 10 ng/mL hSCF (Sigma-Aldrich, GF021), 3 µM PD0325901 (MCE, Monmouth Junction, NJ, USA, HY-10254), 10 µM SB431542 (MCE, HY-10431) and 1% penicillin–streptomycin (Gibco, 15140148). PGCs were isolated from the genital ridge of chicken embryos incubated for 5.5 days and cultured in KO-DMEM containing 10% KSR, 2.0 mM GlutaMax, 1% non-essential amino acids, 0.2% chicken serum, 0.1 mM β-mercaptoethanol, 0.2% ovalbumin (Sigma-Aldrich, A5503), 1.2 mM sodium pyruvate (Gibco, 11360070), 0.01% heparin sodium (MCE, HY-17567A), 25 ng/mL activin A (MCE, HY-P70311), 1% EmbryoMax Nucleosides (Sigma-Aldrich, ES-008) and 1% penicillin–streptomycin. ESCs and PGCs were maintained in a 5% CO2 humidified atmosphere at 37.0 °C. The isolation and culture methods for chicken ESCs and PGCs have been described previously [8].

### 2.3. Cell Induction

One-step induction model: The ESCs that were in good condition were inoculated into the 6-well plates with the induction medium containing KO-DMEM, 10% KSR, 2.0 mM GlutaMax, 1% non-essential amino acids, 1% chicken serum, 0.1 mM β-mercaptoethanol, 1% penicillin–streptomycin, 40 ng/mL BMP4, 40 ng/mL BMP8b and 40 ng/mL EGF.

Two-step induction model: The ESCs that were in good condition were inoculated into the 6-well plates with EB induction medium containing KO-DMEM, 10% KSR, 2.0 mM GlutaMax, 1% non-essential amino acids, 1% chicken serum, 0.1 mM β-mercaptoethanol, 1% penicillin–streptomycin and 10-5 M RA (Sigma-Aldrich, R2625). Half medium was replaced every 2 days. After culturing for 4 days, the same medium was used, in which 10-5 M RA was replaced by 40 ng/mL BMP4, 40 ng/mL BMP8b and 40 ng/mL EGF for PGCLC induction; 4i: CHIR99021 (MCE, HY-10182), PD0325901 (MCE, HY-10254), BIRB796 (MCE, HY-10320), SP600125 (MCE, HY-12041) or Forskolin (MCE, HY-15371), Rolipram (MCE, HY-16900) were added to optimize the PGCLC induction system.

### 2.4. RNA-Seq Data Analysis

Total RNA was extracted from ESCs and PGCs using TRNzol (Tiangen, Beijing, China, DP424) for library construction, quality control [11] and transcriptome sequencing (Shanghai OE Biotechnology, Shanghai, China). The raw data were quality assessed and filtered using FastQC to obtain high quality and relatively accurate valid data [12]. Gene expression (FPKM value) was analyzed using Cuffdiff after comparing the valid data of the samples with the reference genome using HISAT2 [13], and differentially expressed genes were screened using *p* Value < 0.05 & |log2FC| > 2. The process and method of GO [14] and KEGG [15] enrichment analysis for differentially expressed genes were carried out, referring to relevant studies.

### 2.5. qRT-PCR

Total RNA was extracted from DF-1, ESCs, PGCs, EBs and PGCLCs formed in different induction systems using TRNzol and reverse-transcribed into cDNA with HiScript III RT SuperMix (Vazyme, Nanjing, China, R323-01). β-actin was used as an internal control gene to estimate the expression of pluripotency marker genes (Nanog, Lin28, Sox2, Oct4), marker genes of three germ layers (Pax6, Eomes, Vimentin, Sox17) and the germ cell marker (Blimp1, Dazl, Cxcr4, Cvh). The reaction mixture for qRT-PCR included 10 μL 2×ChamQ Universal SYBR qPCR Master Mix, 0.6 μL each of upstream and downstream primer, 2 μL cDNA and 6.8 μL ddH2O. The reaction procedure was performed according to the instructions provided by ChamQ Universal SYBR qPCR Master Mix (Vazyme, Q711-02). The primers are supplied in Appendix A.

### 2.6. AKP Staining

After removing the medium from the plates, the ESCs were washed twice with PBS and fixed with the cell fixing solution at RT for 5 min. Then, the cells were washed twice with PBS and stained with the freshly configured ALP staining solution (Solarbio, G1480). After incubating at RT for 20 min without light, the cells were washed twice and stained with nuclear solid red staining solution (Solarbio, Beijing, China, G1480) for 5 min. The cells were washed twice and observed and photographed under the microscope.

### 2.7. Indirect Immunofluorescence

The collected ESCs and PGCLCs formed in different induction systems were fixed with 4% paraformaldehyde for 30 min, washed with PBS three times and treated with 1% Triton X-100 for 15 min, followed by the addition of 10% FBS to block the cells for 2 h. Then, the primary antibodies against SSEA-1 (Abcam, Cambridge, MA, USA, ab16285), OCT4 (Abcam, ab19857) and CVH (Abcam, ab13840) were added and incubated at 37 °C for 2 h and then at 4 °C overnight. After washing three times with PBST, the cells were incubated with the corresponding secondary antibodies at 37 °C for 2 h without light, and then washed three times with PBST, followed by incubation with DAPI for 10 min without light, and then washed again. Fluorescence was observed and photographed under the fluorescence microscope.

### 2.8. Cell Cycle Assay

A cell cycle assay kit (Beyotime, Shanghai, China, C1052) was used to detect cell cycle changes in different generations of ESCs. The collected ESCs were re-selected with 70% ethanol and fixed at 4 °C for 30 min. After, the cells were resuspended with 70% ethanol and fixed at 4 °C for 30 min. After the cells were collected using centrifugation, 500 μL PI staining solution was added, and it was incubated at 37 °C for 30 min without light. The stained cells were detected using flow cytometry.

### 2.9. Flow Cytometry (FC)

PGCLCs formed in different induction systems were collected and washed with PBS. Next, 1% Triton X-100 was added for 15 min to permeabilize the membrane, and the cells were blocked with 10% FBS for 2 h. CVH antibody was added and incubated at 37 °C for 2 h and then at 4 °C overnight. The secondary antibody was added after washing the cells three times with PBST, and then it was incubated at 37 °C for 2 h in the dark. The suspension was centrifuged, and the supernatant was discarded. The cells were resuspended with PBS and detected using flow cytometry.

### 2.10. Migration Experiment

The cells were stained with the PKH67 staining kit (Solarbio, D0031). When the recipient chicken embryos developed to HH17 (2.5 d), a small hole was made at the blunt end of the egg and a window, approximately 1 cm in diameter, was carefully created with forceps. A total of 3000 stained cells were injected into the dorsal aorta of the recipient embryo under a stereomicroscope and 20 μL of penicillin–streptomycin was dropped into the window. Then, the window was sealed with medical breathable tape, and the eggs were put back into the incubator for incubation. The recipient chicken embryos were collected after developing to HH32 (7.5d) and the migration of donor cells to the gonads was observed under a stereoscopic fluorescence microscope.

### 2.11. Data Analysis

All experiments were repeated at least three times, and the data are presented as mean ± standard error. After all data were sorted using EXCEL, SPSS 19.0 (SPSS, Chicago, IL, USA) and Graph Pad Prism 6 (GraphPad Software Inc., San Diego, CA, USA) were used to perform significance analysis and generate diagrams. Significant differences between the groups were determined through a one-sample *t*-test and one-way ANOVA (* *p* < 0.05, significant difference; ** *p* < 0.01, extremely significant difference).

## 3. Results

### 3.1. High-Quality cESCs Were Obtained for the Induction of cPGCs In Vitro

To obtain high-quality ESCs for the subsequent PGC induction experiment, the primary chicken ESCs were cultured in the “2i+LIF” system in vitro, passaged and identified. During culturing in the “2i+LIF” system, the results of cell morphological observations showed that chicken ESCs were nest-like with clear boundaries and that there were no differentiated cells at the edges of the clones, within which the ESCs were densely arranged, making it difficult to distinguish single cells. The ESCs could make passage to the 8th-generation stably (Figure 1A). The cultured ESCs were positive for alkaline phosphatase (AKP), SSEA-1 and OCT4 (Figure 1B,C). Results from the qRT-PCR showed that the expression of pluripotency marker genes (Oct4, Sox2, Lin28, Nanog) was significantly higher in the ESCs than in the PGCs (*p* < 0.01) (Figure 1D). In addition, there were no significant differences in the growth rate (Figure 1E) or cell cycles (Figure 1F and Appendix A) of the ESCs from the different generations (P2 and P8) (*p* > 0.05). These results indicate that the isolated chicken ESCs could be utilized in subsequent experiments.

### 3.2. cESCs Were Induced into Embryoid Bodies by RA

In order to establish a two-step PGC induction model, this study intended to use retinoic acid (RA) (10^−5^ M) [7] to induce ESCs into embryoid bodies (EBs, Figure 2A). The results of morphological statistics showed that cells began to differentiate and formed a dense internal structure at day 2–day 3 after RA induction, at which time EBs had formed. The number of EBs continued to increase (Figure 2B and Appendix A). To further evaluate the EB formation process, we examined the relative expression of the marker genes of three germ layers, *Vimentin* (for endoderm), *Eomes* (for mesoderm) and *Pax6* (for ectoderm), during induction. The results of the qRT-PCR showed that *Vimentin* and *Eomes* were consistently upregulated throughout the induction of EBs, while the expression of *Pax6* started to decrease at day 2 but showed an overall increasing trend, possibly because, during the EB formation, the ectoderm formed first and subsequently differentiated to form the mesoderm and endoderm (Figure 2C). Further gene expression analysis showed that the expressions of *Vimentin*, *Eomes* and *Pax6* were higher in EBs than in ESCs and PGCs (Appendix A). These results indicate that ESCs were induced to differentiate into EBs with three germ layers after 4 days of culturing with RA.

### 3.3. BMP4/BMP8b/EGF Can Induce EBs into PGCLC

It is universally acknowledged that BMP4, BMP8b, EGF and other cytokines are involved in regulating the formation and proliferation of PGCs. In order to efficiently induce EBs into PGCLCs, RA was replaced with BMP4/BMP8b/EGF at day 4 after RA induction in this study (Figure 3A). Morphological observations showed that at day 5 and day 6, EBs became larger and began to break; at day 7, no complete EBs could be observed in the field of view and EBs started to disintegrate into small round cells; at day 8, clusters of PGC-like cells (PGCLCs) were observed around the disintegrated EBs (Figure 3B). The results of the qRT-PCR showed that the expression of reproductive marker genes (*Cxcr4*, *Cvh*, *Dazl*, *Blimp1*) was upregulated (Figure 3C), while the expression of the marker genes of three germ layers was downregulated (Figure 3D) during PGCLC induction, which preliminarily indicated that EBs were induced into PGCLCs by BMP4/BMP8b/EGF. Moreover, an indirect immunofluorescence assay confirmed that the cells expressed PGC marker protein CVH after 4 days of inducing EBs to differentiate (Figure 3E). Flow cytometry analysis revealed that the PGCLC formation efficiency was 15.8% ± 0.4%, which was significantly higher than that of the one-step induction model (Figure 3F). In conclusion, EBs were induced into PGCLC by BMP4/BMP8b/EGF, and a two-step induction model of chicken PGCs in vitro was successfully established.

### 3.4. Similar Characteristics of PGCs Formation between Chicken and Other Species Were Identified through RNA-Seq

In order to systematically analyze the key factors affecting the formation of chicken PGCs and further optimize the in vitro induction system of chicken PGCs, purified ESCs and PGCs were collected, and transcriptome sequencing was performed in this study. A total of 5025 genes were screened and identified as differentially expressed genes (DEGs) (*p* Value < 0.05 & |log2FC| > 2); of these, 3255 genes were upregulated and 1770 genes were downregulated (Figure 4A–D). Pearson’s correlation analysis and the PCA analysis of transcriptomic results revealed obvious differences between ESCs and PGCs (Figure 4A and Appendix A). Among the DEGs, the expression of genes related to pluripotency, *Oct4*, *Lin28* and *Nanog*, were significantly higher in ESCs than in PGCs, while the expression of reproductive marker genes *BMP4* and *CDH4* and the key transcription factor *TFCP2L1* were significantly higher in PGCs than in PGCs (Figure 4D). These results were consistent with the basic characteristics of the samples, indicating that the transcriptome sequencing results were accurate and repeatable and could be further analyzed.

To clarify the differences between ESCs and PGCs, GO functional annotation was performed on the DEGs (*p* Value < 0.05 & ListHits > 3). The results showed that these DEGs were enriched to a total of 644 GO terms (Appendix A). Among these GO terms, terms related to stem cell differentiation, germ cell development and cell migration were significantly enriched. In stem-cell-differentiation-related terms, the expression of *Sox17*, *Nanog*, *Oct4* and *LIF* genes were significantly downregulated during the PGC formation, while in germ-cell-development- and cell-migration-related terms the expressions of *Cftr*, *Nanos1*, *Tgfb1* and *Lhx1* were significantly different (Figure 4E–G; Appendix A).

Meanwhile, we analyzed the signaling pathways enriched by the DEGs. The results of GO analysis showed that 115 genes were enriched in 33 GO terms related to the Wnt signaling pathway; 63 genes were enriched in 11 GO terms related to the BMP4 signaling pathway; and 51 genes were enriched in 12 GO terms related to the Notch signaling pathway (Appendix A). The analysis of the expression of key genes in the signaling pathways showed that these signaling pathways were significantly activated during the formation of chicken PGCs (Appendix A). The results of KEGG enrichment analysis were consistent with those of GO analysis (Appendix A). These results shared some similarities with the developmental process of PGCs in species such as mammals and *Drosophila*. Therefore, we concluded that the developmental process of chicken PGCs was partially similar to that of other species.

### 3.5. New Factors Can Be Used to Optimize PGCs’ Induction Models In Vitro

It is a commonly accepted fact that BMP4 in a BMP signal extensively regulates the origin process of PGCs in different species, despite differences in how and where PGCs originate among these species. Importantly, the BMP signal is involved in a large number of interactions with Wnt or Notch signals during BMP signal functioning (Appendix A). For example, a Wnt signal strengthens the regulation of a BMP signal on reproductive marker gene *Blimp1*, and Wnt and Notch signals coordinate to regulate the proliferation ability of PGCs in different species during migration (Appendix A). In the established two-step PGC induction model, we used BMP4, BMP8b, EGF and other cytokines to activate the BMP signaling pathway, but the PGCLC induction efficiency in vitro was not satisfactory. Therefore, new factors needed to be sought to optimize the PGC induction model. The results of GO and KEGG pathway enrichment analyses showed that MAPK and cAMP were significantly enriched (Appendix A). Dual-specificity phosphatases (DUSPs), the endogenous inhibitors of the MAPK signaling pathway, were enriched in this signaling pathway as DEGs. Except for DUSP4, the expressions of DUSP5, DUSP10, DUSP1, DUSP8 and DUSP6 were all upregulated in PGCs. DUSP4 is a negative cell cycle regulator, mainly regulating the G2/M phase of mitosis [16], and its expression is significantly higher in ESCs than in PGCs (*p* < 0.01). The results of the dynamic expression of these genes indicated that the inhibition of the MAPK signaling pathway could promote the differentiation of ESCs and the formation of PGCs (Appendix A). In addition, most of the DEGs enriched in the cAMP signaling pathway (PKD2, RAP1A, WT1, EZR, SLC26A6, etc.) were significantly upregulated in PGCs, indicating that the activation of the cAMP signal contributed to PGC formation (Appendix A). Interestingly, when we conducted a protein–protein interaction analysis, we found that the BMP4 signal had little interaction with cAMP and MAPK signals (Appendix A), indicating that cAMP and MAPK signaling pathways can independently participate in the regulation of PGC formation. Therefore, in this study, the activation or inhibition of these signaling pathways was intended to optimize the PGC induction model in vitro.

### 3.6. Optimization of Chicken PGCs’ Two-Step Induction Model

In order to improve the PGC induction efficiency in vitro, the inhibitors CHIR99021, PD0325901, BIRB796 and SP600125 (4i) were used to inhibit GSK-3β (activating the Wnt signal) and the MAPK signaling pathway, while Forskolin and Rolipram were used to activate the cAMP signaling pathway (Figure 5A). Morphological observations showed that, after the inhibition of GSK, MEK, MAPK and JNK signals by 4i and the activation of the cAMP signal by Forskolin and Rolipram, EBs became larger, began to break and released PGCLCs (Figure 5B). The results of the qRT-PCR showed that the expression of PGC marker genes *Cxcr4*, *Cvh*, *Dazl* and *Blimp1* displayed an increasing trend during PGCLC formation (Figure 5C). Meanwhile, the indirect immunofluorescence detection of cells showed that PGCLCs induced with 4i or Forskolin and Rolipram all expressed PGC marker protein CVH, suggesting that the chicken PGCLC formed after the optimization of the inducing system also possessed the same biological characteristics of PGCs (Figure 5B). Furthermore, flow cytometry analysis demonstrated that after adding 4i, the proportion of CVH+ PGCLCs was 29.5% ± 2.2; meanwhile, after adding Forskolin and Rolipram, the proportion of CVH+ PGCLCs reached 33.8% ± 0.8%; both of these results were significantly higher than the control group (*p* < 0.05) (Figure 5D), which was consistent with the quantitative results. The migration experiment also showed that the induced PGCLCs could migrate to the gonads in the recipient chicken embryos, although the migration efficiency was lower than that of the isolated PGCs (Appendix A). In summary, the PGC induction efficiency was successfully increased to about 33.8% in vitro in this study by small-molecule compounds.

## 4. Discussion

Here, by using RNA-seq, we found that the PGC formation process of chicken shares some similarities with that of other species; that is, the formation process is regulated by signals such as BMP, MAPK and cAMP, based on which we successfully established and optimized a two-step induction model for chicken PGCs in vitro and significantly improved the efficiency of PGC formation in vitro.

It is well known that PGCs are differentiated from ESCs but are not directly derived from ESCs. During normal embryonic development, the zygote forms a blastocyst through cleavage, at which time the cells are developmentally pluripotent, and the inner cell mass from the blastocyst are ESCs [17]. The blastocyst develops into a gastrula and forms three germ layers to develop into different cells, tissues and organs of the body during further development. Despite the origin of PGCs varying among different species, the embryonic development of different species follows this rule [1]. In the process of establishing a PGCs induction model in vitro, we all paid attention to the fact that intermediate cell types, generally embryoid bodies or extra-embryonic ectoderm, appear during the differentiation from pluripotent stem cells into PGCLCs. Further studies showed that increasing the formation efficiency of embryoid bodies or extra-embryonic ectoderm can significantly promote the formation of PGCLCs. Hayashi et al. [6] found that inducing mEpiLCs from mESCs into mPGCLCs significantly improves the mPGCLC induction efficiency. In the study by Irie et al. [18], hESCs were induced into EBs before hPGCLC induction, and the induction efficiency reached 28.9%. von Meyenn et al. [19] induced hEpiLCs from hESCs into hPGCLCs with an induction efficiency of 25.9%. In our study, it was found that the EB formation efficiency in the conventional one-step induction model was low, which greatly affects the PGCLC formation efficiency, but the PGCLC formation efficiency was improved by RA.

The function of canonical Wnt signaling relies on β-catenin to enter the nucleus. The key signaling molecule GSK-3β can inhibit the phosphorylation of β-catenin so that β-catenin will not be degraded [20]. Yin et al. [21] found that CHIR99021 can be used to inhibit GSK-3β and activate the Wnt signal during the study of the development of pig embryos. Wang et al. [22] also used CHIR99021 to activate the Wnt signal after inhibiting GSK-3β to promote osteogenesis in mice. Studies have shown that Wnt signaling is essential in the development of PGCs in other species. In *Drosophila*, Wnt signaling mediates lipid metabolism to regulate the migration of PGCs [23]. Shono et al. [24] found that the timely activation and inhibition of Wnt signaling is very important for the differentiation of PGCs in marmosets. Kimura et al. [25] found that Wnt signaling regulates the G1/S transition in proliferating mouse PGCs. The MAPK signaling pathway is a key signaling pathway that regulates cell proliferation and apoptosis. Numerous studies have shown that MAPK signaling plays an important role in the maintenance of stem cell pluripotency, as well as in tumor development [26,27]. Three branches, ERK/MAPK [28], JNK/MAPK [29] and p38/MAPK [30], are included in canonical MAPK signaling, which is involved in a variety of cellular developmental processes, especially in the regulation of cell differentiation. Ding et al. [28] found that ERK/MAPK reduces the proliferation rate of mouse neural stem cells while inhibiting their differentiation to neurons. In the study by Yang et al. [29], JNK/MAPK was confirmed to regulate the differentiation of human periodontal ligament stem cells to osteoblasts and adipocytes. Li et al. [30] found that p38/MAPK is essential for the differentiation of mouse osteoclast precursors into osteoclasts. In our study, we utilized the small-molecule compound 4i [31,32] to activate Wnt signaling and inhibit MAPK signaling, significantly improving the PGC formation efficiency in vitro.

The cAMP signal is an important signal regulating cell development, and can increase the expression of TβRI and coordinate with TGF-β to regulate the proliferation of chicken germ cells. Moreover, the cAMP signal can enhance the expression of Smad3 and promote its phosphorylation to improve its activity, participating in the regulation of TβRII on germ cells [33]. In other words, an obvious interaction exists between the cAMP signal and BMP4 signal. It is well established that the BMP4 signal is critical for PGC development, so it can be inferred that the cAMP signal is involved in PGC formation. Forskolin and Rolipram are important activators of cAMP signaling, mainly by inhibiting phosphodiesterase PDE4 to increase cAMP concentration [34,35]. Ohta et al. [36] have demonstrated that Forskolin and Rolipram are able to increase intracellular cAMP concentration through a synergistic mechanism, thus promoting PGC formation. In our study, Forskolin and Rolipram were used to effectively optimize the in vitro chicken PGC induction system, and the induction efficiency reached 33.8% ± 0.8%.

That the PGCLCs induced possess the germline transmission ability is the key index to evaluate the success of the whole induction system. The 4i and Forskolin/Rolipram used in our study mainly function by activating/inhibiting Wnt, MAPK, and cAMP signaling pathways which have all been reported to regulate PGC migration. Chen et al. [37] found that CST1 promoted gastric cancer cell migration by activating Wnt signaling pathway. SAA was confirmed to inhibit astrocyte migration by activating p38/MAPK [38]. In the study by Kim et al. [39], cAMP was shown to enhance mESC migration by coordinating the disruption of cellular adhesion junctions and the rearrangement of cytoskeletal proteins. These are consistent with the results of our migration experiment that the PGCLCs formed in the induction system have the germline transmission ability.

## 5. Conclusions

By analyzing the difference between chicken ESCs and PGCs, we screened out signaling pathways such as cAMP, MAPK and Wnt that are involved in the formation of PGCs, and found that, during the formation of PGCs, the cAMP and Wnt signaling pathways were activated, while the MAPK signaling pathways were inhibited. At the same time, we successfully established a two-step induction model for chicken PGCs to improve the induction efficiency of PGCs in vitro by adding activators or inhibitors of different signaling pathways.

## Figures and Tables

**Figure 1 animals-14-00302-f001:**
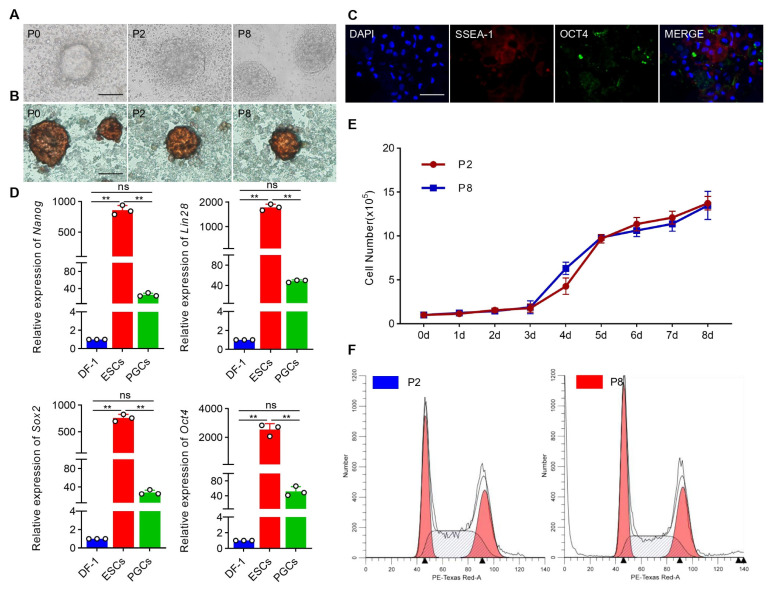
High-quality cESCs were obtained for the induction of cPGCs in vitro. (**A**) Morphological observation of ESCs from different generations. Scale bar: 60 µm. (**B**) ESCs clones (brownish-red) were stained with AKP staining. Scale bar: 60 µm. (**C**) The cultured ESCs were identified through indirect immunofluorescence detection with SSEA-1 and OCT4 antibody. Scale bar: 60 µm. (**D**) qRT-PCR was used to detect the expression of pluripotency marker genes (Nanog, Lin28, Sox2, Oct4) in ESCs and PGCs (data are shown as mean ± SEM, n = 3 independent experiments, ** *p* < 0.01, one-way ANOVA). DF-1 cells were used as control. (**E**) Proliferation curves of ESCs (P2 and P8) (data are shown as mean ± SEM, *n* = 3 independent experiments, ** *p* < 0.01, *t* test). (**F**) Cell cycle analysis of ESCs (P2 and P8).

**Figure 2 animals-14-00302-f002:**
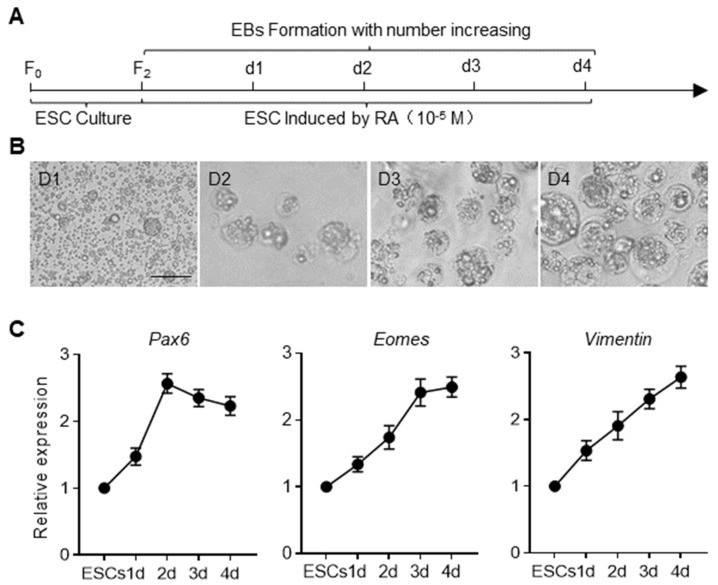
cESCs were induced into EBs by RA. (**A**) Schematic diagram of ESCs being induced into EBs in vitro. (**B**) Morphological observation of EBs during induction. Scale bar: 60 µm. (**C**) qRT−PCR was used to detect the expressions of marker genes of three germ layers, Pax6 (ectoderm), Eomes (mesoderm) and Viment (endoderm), in EBs during induction.

**Figure 3 animals-14-00302-f003:**
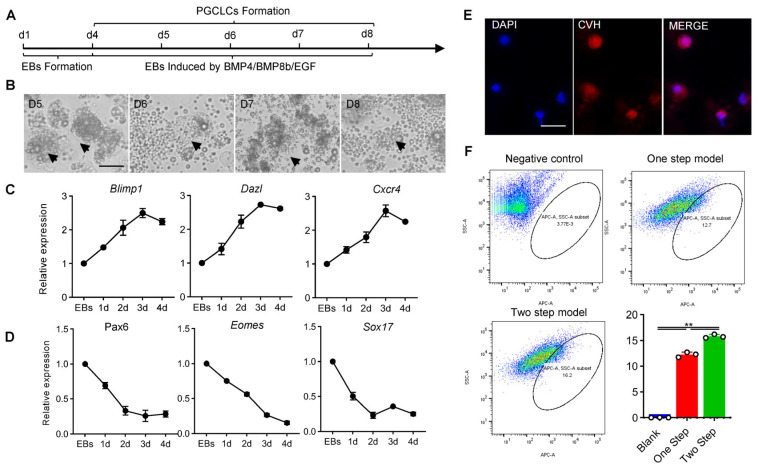
BMP4/BMP8b/EGF can induce EBs into PGCLC. (**A**) Schematic diagram of inducing EBs into PGCLCs in vitro. (**B**) Morphological observation of the formation of PGCLCs during induction. The arrows indicate PGCLCs. Scale bar: 60 µm. (**C**) qRT-PCR was used to detect the expression of reproductive-related marker genes (Cxcr4, Cvh, Dazl, Blimp1) and in PGCLCs during induction. (**D**) qRT-PCR was used to detect the expression of marker genes of three germ layers, Pax6 (ectoderm), Eomes (mesoderm) and Sox17 (endoderm), in PGCLCs during induction. (**E**) Indirect immunofluorescence detection with CVH marker in PGCLCs. Scale bar: 60 µm. (**F**) Detection of PGCLC formation efficiency through flow cytometry using the CVH antibody (data are shown as mean ± SEM, n = 3 independent experiments, ** *p* < 0.01, one-way ANOVA).

**Figure 4 animals-14-00302-f004:**
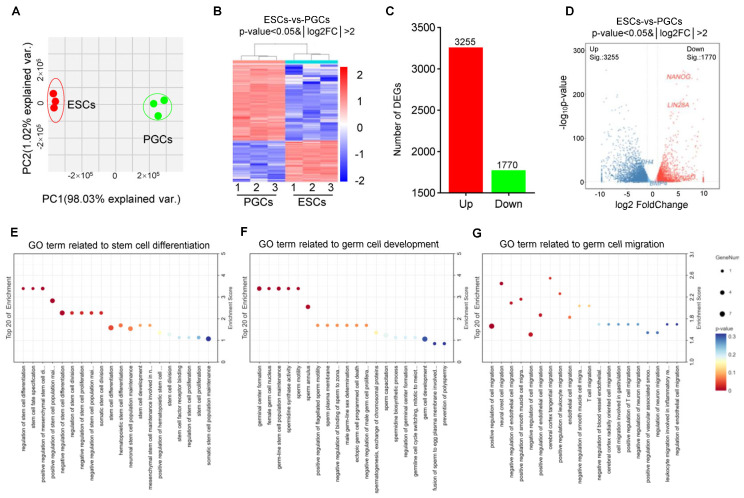
Differences between ESCs and PGCs were analyzed using RNA−seq. (**A**) PCA was used to analyze the differences between 3 samples of ESCs and PGCs. (**B**,**C**) Statistics of differentially expressed genes during PGCs’ formation. (**D**) Volcanic map of differential genes between ESCs and PGCs. (**E**–**G**) GO analyzed the enrichment of differentially expressed genes in related items of “stem cell differentiation”, “germ cell development” and “germ cell migration”.

**Figure 5 animals-14-00302-f005:**
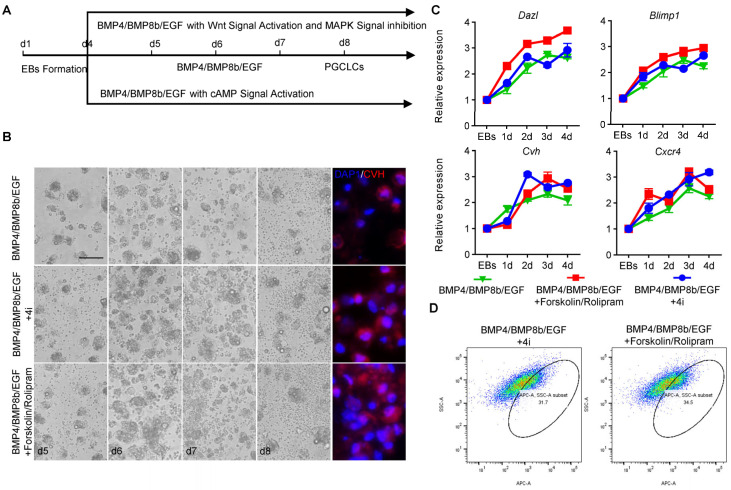
Optimization of chicken PGCs’ two-step induction model with 4i or Forskolin/Rolipram. (**A**) Schematic diagram of EBs induced into PGCLCs with different treatments in vitro. (**B**) Morphological observation of the formation of PGCLCs during induction with different treatments (left). Indirect immunofluorescence detection of the efficiency of the formation of PGCLCs at day 4 after induction with different treatments (right). Scale bar: 60 µm. (**C**) qRT-PCR was used to detect the expression of PGC marker genes (*Cxcr4*, *Cvh*, *Dazl*, *Blimp1*) in PGCLCs during induction with different treatments (data are shown as mean ± SEM, n = 3 independent experiments, one-way ANOVA). (**D**) Detection of PGCLC formation efficiency after induction with different treatments by flow cytometry using the CVH antibody.

## Data Availability

Data and codes related to this paper may be requested from the authors. The data of RNA-seq for ESC and PGC in this paper have been deposited in the GEO database with the Accession No. GSE159511.

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
