# Peer review of "The Establishment and Optimization of a Chicken Primordial Germ Cell Induction Model Using Small-Molecule Compounds"

_animals, 2024, doi:10.3390/ani14020302_

Round 1
Reviewer 1 Report
Comments and Suggestions for Authors
It is an interesting study and an endeavor to induce ES cells to form Avian PGCs. It is not sufficiently illustrated in the manuscript that it is a constant argument with mammalian scientists that avian ES cells are ES cells because Avian cells undergo a germ-line fate with low efficency.
The presentation of the study is confusing, there is a comparison of ES cells and PGCs, which is fine, and then there is an induction protocol that is difficult to follow. It is clear at one point that the authors were trying to make ES cells into PGCs
However, the authors should firstly prove their ES cells are ES cells through chorioallantoic grafting. In my view, it is a small flaw in the study.
However, the major flaw in the study is that the authors have failed to prove that their purported PGCs are actually PGCs. The authors must inject their PGC cell lines back into the blastoderm or dorsal aorta and show migration to the gonad. Secondly, the authors must show that the gene in their PGCs is carried from one generation of birds and the next generation of birds. In other words, the authors MUST clearly demonstrate germline chimerism to demonstrate PGC function. There are a variety of methods the authors can use even classical feather markers but the present work does nothing to prove they have induced a PGC phenotype without supporting the molecular work with a functional assay (i.e. birds that are clearly germline Chimeras).
Comments on the Quality of English LanguagePlease have the manuscript edited by a qualified editor. First person is not appropriate for scientific writing. Repeated un-necessary phrases such as In this study, in fact, in particular, in order to. Please consult Strunk and White the Elements of Style.
Author Response
(1) The manuscript has been improved and Reviewer comments were properly addressed. However, Authors included results of the migration experiment mentioned by Reviewer 1 but there is no description of this experiment in M&M section and results were not discussed. Since this is an important (according to Reviewer 1) part of the data which is crucial for conclusions, both description of migration experiment and comments on the obtained data should be included in the manuscript.
Response:Thank you for your comments. First of all, we have added the description of the migration experiment in M&M section. For details, see line 165-173:
2.10 Migration experiment
The cells were stained with the PKH67 staining kit (Solarbio, D0031). When the recipient chicken embryos developed to HH17 (2.5 d), a small hole was made at the blunt end of the egg and a window, approximately 1 cm in diameter was carefully created with forceps. 3000 stained cells were injected into the dorsal aorta of the recipient embryo under a stereomicroscope and 20 μL of penicillin–streptomycin was dropped into the window. Then the window was sealed with medical breathable tape, and the eggs were put back to the incubator for incubation. The recipient chicken embryos were collected after developing to HH32 (7.5d) and the migration of donor cells to the gonads was observed under a stereoscopic fluorescence microscope.
Secondly, we have also discussed this phenomenon appropriately in the discussion section. For details, see line 395-403:
That the PGCLCs induced possess the germline transmission ability is the key index to evaluate the success of the whole induction system. The 4i and Forskolin/Rolipram used in our study mainly function by activating/inhibiting Wnt, MAPK, and cAMP signaling pathways which have all been reported to regulate PGC migration. Chen et al. [37] found that CST1 promoted gastric cancer cell migration by activating Wnt signaling pathway. SAA was confirmed to inhibit astrocyte migration by activating p38/MAPK [38]. In the study by Kim et al. [39], cAMP was shown to enhance mESC migration by coordinating the disruption of cellular adhesion junctions and the rearrangement of cytoskeletal proteins. These are consistent with the results of our migration experiment that the PGCLCs formed in the induction system have the germline transmission ability.

Reviewer 2 Report
Comments and Suggestions for Authors
Simple Summary: no comments
Abstract: In line 16, authors can mention primordial germ cell (PGC), as they did in introduction. line 21, full form of ESC. Same goes with line 22, DEG.
Introduction: In line 35, "more and more attention has been paid to the prospects ", this is not great for a first line of introduction. Authors can consider to rephrase it.
In line 48, full form of PGCLCs is missing.
Materials and Methods: in line 150, one place "30min", in another place "30 min". It can be either way but should be same for the whole manuscript. Please check for this small things throughout the manuscript.
Results: Well presented results. No comments here.
Discussion: Bit longer, can consider concise little bit.
Conclusions: no comments
Comments on the Quality of English Language
No comments
Author Response
Reviewer 2
Reviewer 2-1 Abstract: In line 16, authors can mention primordial germ cell (PGC), as they did in introduction. line 21, full form of ESC. Same goes with line 22, DEG.
Response:Thank you for your comments. They have been revised in line 16, 22, 23.
Reviewer 2-2 Introduction: In line 35, "more and more attention has been paid to the prospects ", this is not great for a first line of introduction. Authors can consider to rephrase it.
Response:Thank you for your comments. It has been revised as ‘In recent years, primordial germ cells (PGCs) have been widely used in the fields of genetic engineering, regenerative medicine and germplasm conservation due to its unique biological characteristics.’ in line 35-37.
Reviewer 2-3 In line 48, full form of PGCLCs is missing.
Response:Thank you for your comments. PGCLC is short for PGC-like cell. It has been revised in line 21.
Reviewer 2-4 Materials and Methods: in line 150, one place "30min", in another place "30 min". It can be either way but should be same for the whole manuscript. Please check for these small things throughout the manuscript.
Response:Thank you for your comments. We have checked and revised the entire manuscript.
Reviewer 2-5 Discussion: Bit longer, can consider concise little bit.
Response:Thank you for your comments. We have revised the discussion appropriately.
Reviewer 3 Report
Comments and Suggestions for Authors
Dear authors,
your submitting manuscript is remarkable and of a high standard, but I have a few comments:
Abstract must be shorter (according to the instructions for author)
Line 16-17, 57, - "in vitro" - must be in italic
Author Response
Reviewer3
Reviewer 3-1 Abstract must be shorter (according to the instructions for author)
Response:Thank you for your comments. It has been revised.
Reviewer 3-2 Line 16-17, 57, - "in vitro" - must be in italic
Response:Thank you for your comments. They have been revised.